



# Downscaling Atmospheric Chemistry Simulations with Physically Consistent Deep Learning

Andrew Geiss[1], Sam J. Silva[1,2], and Joseph C. Hardin[1,3]

[1]Pacific Northwest National Laboratory, Richland, WA, USA
[2]University of Southern California, Los Angeles, CA, USA
[3]ClimateAi, Inc. San Francisco, CA, USA

**Correspondence:** Andrew Geiss (andrew.geiss@pnnl.gov)

**Abstract.** Recent advances in deep convolutional neural network (CNN) based super resolution can be used to downscale atmospheric chemistry simulations with substantially higher accuracy than conventional downscaling methods. This work both demonstrates the downscaling capabilities of modern CNN-based single image super resolution and video super resolution schemes and develops modifications to these schemes to ensure they are appropriate for use with physical science data. The

CNN-based video super resolution schemes in particular incur only 39% to 54% of the grid-cell-level error of interpolation schemes and generate outputs with extremely realistic small-scale variability based on multiple perceptual quality metrics while performing a large ($8 \times 10$) increase in resolution in the spatial dimensions. Methods are introduced to strictly enforce physical conservation laws within CNNs, perform large and asymmetric resolution changes between common model grid resolutions, account for non-uniform grid cell areas, super resolve log-normally distributed datasets, and leverage additional inputs such

as high-resolution climatologies and model state variables. High-resolution chemistry simulations are critical for modeling regional air quality and for understanding future climate, and CNN-based downscaling has the potential to generate these high resolution simulations and ensembles at a fraction of the computational cost.

## 1   Introduction

The chemical composition of the atmosphere is tightly coupled to many important processes in the global Earth System,

including air pollution exposure, biogeochemical cycles, and the Earth's radiative budget. Exposure to atmospheric pollution, much of which is formed through chemical reactions in the atmosphere, is the leading environmental cause of death worldwide, responsible for millions of premature deaths per year (Forouzanfar et al., 2015). Global biogeochemical cycles are strongly modulated by the chemical composition of the atmosphere, including effects of greenhouse gases, aerosols, and toxic pollutants (e.g. Clifton et al. (2020); Mahowald (2011)). The impact of atmospheric composition on Earth's radiative budget is a major

driver of modern climate change through both direct absorption and scattering of radiation and indirect interactions with a variety other radiatively important processes (e.g. aerosol-cloud interactions, etc.) (Committee on the Future of Atmospheric Chemistry Research et al., 2016).

Many of these globally relevant processes are fundamentally controlled at very small length scales, motivating the development of high-resolution computational model representations of atmospheric chemistry over large spatial domains (e.g.





continental-global). These models solve the continuity equation for atmospheric chemical constituents, capturing the relevant known physical and chemical processes to allow for prediction at fine spatial resolution. These fine scale simulations enable scientific and policy-relevant insights that are not possible with coarser model predictions (e.g. Hu et al. (2018); Keller et al. (2021)). However, the large computational expense of running these high-resolution models can be a limiting factor in their adoption and application.

To address this issue of computational expense, a variety of post-processing techniques have been developed for predicting atmospheric composition at high resolution. These range from simple statistical scaling to advanced machine learning architectures. For example, Geddes et al. (2016) scale coarser observed maps of nitrogen dioxide by the spatial distribution observed from a higher resolution instrument, which is ultimately used to infer long term trends in NO2 concentrations. Recent work by Sun et al. (2021) use a Bayesian Neural Network to combine a variety of data sources for high resolution surface ozone concen-
tration predictions, allowing for improved understanding in long term ozone trends. These studies on atmospheric composition and related research across the earth sciences (e.g. Anh et al. (2019); Bedia et al. (2019); Vandal et al. (2018)) demonstrate the value in these post-processing downscaling approaches as a computationally efficient technique for high resolution prediction.

## 1.1 Convolutional Neural Networks and Super Resolution

In the last ten years, deep learning research has rapidly expanded. Deep Convolutional Neural Networks (CNNs) have shown
impressive performance improvements over conventional methods for many image processing tasks such as classification (Krizhevsky et al., 2012), object detection (Girshick et al., 2013), and segmentation (Ronneberger et al., 2015). Because CNNs learn the weights of convolutional kernels, rather than processing individual input pixels separately, their learned representations are translationally invariant and they can efficiently represent common 2-D features in their training data. While the majority of CNN research has focused on image processing, they are particularly powerful for processing most data that are
organized on a regular grid.

CNNs have shown impressive results when applied to Single Image Super Resolution (SISR) (Wang et al., 2020). SISR artificially enhances the resolution of images after they are captured. This can easily be accomplished using 2-D interpolation, which estimates sub-pixel data based on neighboring pixels, but the resulting images are often low quality. More sophisticated SISR schemes exist (Nasrollahi and Moeslund, 2014), like "A+" for instance, which can incorporate information from a wider
area surrounding a pixel by comparison to a dictionary of exemplars (Timofte et al., 2015). Recent CNN-based methods can produce even sharper super-resolved imagery however. Initially, Dong et al. (2016) used a 3-layer CNN to achieve state-of-the-art SISR results, and the approach was quickly improved upon with a deeper CNN by Kim et al. (2016). Since then, SISR-CNN architectures have undergone rapid development, trending towards larger and more complex designs. Some key developments have been: incorporation of residual blocks (He et al., 2016; Lim et al., 2017), dense blocks (Huang et al., 2017;
Zhang et al., 2018), and channel attention blocks (Bastidas and Tang, 2019; Zhang et al., 2018) into the architectures; use of transposed convolutions (Long et al., 2014) and pixel-shuffle (Shi et al., 2016) for upsampling[1]; and the use of feature-loss

---

[1]A point of clarification: both the terms "upsampling" and "downscaling" refer to increases in resolution. "Upsampling" is often used in the context of image processing and "downscaling" is used in the context of atmospheric modeling





and adversarial-loss to hallucinate plausible sub-pixel features (Goodfellow et al., 2014; Ledig et al., 2017). State-of-the-art CNN-based schemes can now produce incredibly high fidelity images from very low-quality inputs.

There have been similar advances in CNN-based video super resolution (VSR). The VSR problem is an extension of SISR
where video frames preceding and following an image are used as additional inputs. While the core CNN structures used in many VSR schemes are similar to SISR CNNs, VSR involves several key considerations that SISR does not (Liu et al., 2020). A major component of many VSR schemes is a frame alignment pre-processing step to compensate for camera motions, though this is not a necessary consideration for application to outputs from atmospheric models. VSR schemes often include motion vectors as an additional input, which are often computed using optical flow, and can provide additional skill. Perhaps most
importantly, there are several different approaches to incorporating temporal information in the CNN architectures: either by treating the time-dimension like an extra spatial dimension and using 3D convolutions (Kim et al., 2019), providing separate frames as input channels to 2D convolutions (Yan et al., 2019), or using a recurrent CNN (Haris et al., 2019). The addition of this temporal information in super-resolution schemes can improve performance far beyond what SISR CNNs are capable of.

## 1.2 CNNs for Downscaling Atmospheric Data

The immense cost and high societal impact of atmospheric modeling and observing systems make recent CNN-based super resolution techniques an exciting development with the prospect of enhancing the resolution of atmospheric data at relatively low cost. Several authors have already demonstrated their potential in earth science related applications. Super resolution CNNs have been applied to radar data (Geiss and Hardin, 2020a), wind and solar modeling (Stengel et al., 2020), satellite remote sensing (Liebel and Körner, 2016; Lanaras et al., 2018; Müller et al., 2020), precipitation modeling (Wang et al., 2021),
and climate modeling (Vandal et al., 2018; Baño Medina et al., 2020).

While CNN-based image super-resolution has far outpaced conventional methods in terms of image quality, there are additional considerations when applying these schemes to physical science data: enforcement of known physical laws, multi-modal or multi-resolutional inputs, non-normally distributed data, and irregular grid-spacing to name a few. Of particular importance is the fact that SISR-CNNs do not explicitly enforce consistency between their inputs and outputs. This is problematic if the
schemes are to be applied to scientific data where we may wish to enforce strict agreement between the low-resolution and super-resolved data based on the known properties of the underlying physical system. Several studies have addressed this problem by adding terms to the neural network's loss function that nudge it towards better agreement between the low- and high-resolution data (Ulyanov et al., 2018; Abdal et al., 2019; Menon et al., 2020). Geiss and Hardin (2020b) introduced a method to strictly enforce agreement across resolutions under 2-D averaging. Developing CNNs with internal representations
of known physical properties of the underlying system has been identified as a key hurdle before their potential can be fully realized on problems in the physical sciences (Reichstein et al., 2019; Boukabara et al., 2021; von Rueden et al., 2021; Beucler et al., 2021).



### 1.3 Contributions and Impacts

This work develops and evaluates the techniques and CNN components necessary to apply the impressive super-resolution capabilities of CNNs to the problem of downscaling atmospheric chemistry simulations by incorporating components into a CNN that: strictly and exactly enforce conservation of chemical concentrations between the high-resolution output and low-resolution input, allow the CNN to train on log-normally distributed data, account for the irregular grid cell areas in the atmospheric chemistry model's lat-lon grid, apply different resolution changes along the latitude and longitude dimensions, incorporate high-resolution chemical climatologies, and leverage model state variables as additional inputs. Finally, we incorporate the time-evolution of atmospheric data into the super-resolution process using a CNN-based VSR scheme. The end result is a downscaling CNN that can dramatically outperform conventional downscaling schemes and can guarantee that its outputs remain physically consistent with the input.

This sort of post-processing technique has myriad use cases, all taking advantage of the fact that only one set of computationally expensive high resolution simulations need to completed. Following the training of a CNN for downscaling, a user can generate coarse simulations at relatively low computational cost, and then apply the downscaler to explore the potential high resolution characteristics in that simulation. Model ensembles are an ideal use case for this application, where a large number of high resolution simulations is too resource intensive to complete. Using a downscaling CNN, like the one described in this work, would allow for the majority of ensemble member simulations to be completed at a coarser model resolution, and consequently, at much lower computational expense.

### 2 Data

Here, we use model data available from the NASA GEOS Composition Forecast (GEOS-CF) system to explore the application of CNN downscaling to atmospheric composition (Knowland et al., 2020). The GEOS-CF system predicts the abundance and distribution of a variety of atmospheric chemical species using the NASA GEOS model coupled to the GEOS-Chem chemical transport model (see Keller et al. (2021) for additional information about GEOS-CF). The GEOS-CF simulation output are available on a $0.25 \times 0.25$ degree mesh and we train the CNN to downscale data that have been degraded to $2.0 \times 2.5$ degree resolution. This is an unusual ($8 \times 10$) resolution increase compared to most of the SISR literature because it is both very large and asymmetric, but these are two commonly used model grid resolutions. We use the meteorological replay simulation ("das" files) which uses assimilated meteorology to drive model dynamics (Orbe et al., 2017) for the years 2018-2021, and focus on five well studied atmospheric pollutants: $NO_2$, $SO_2$, $CO$, $O_3$, and $PM_{2.5}$. Each of these compounds has a different source profile, spatial distribution, and atmospheric lifetime, allowing for evaluating the CNN downscaling method in a variety of contexts.



## 3 Method

### 3.1 Neural Network

This study uses the "Enhanced Deep Residual Network" (EDRN) architecture (Lim et al., 2017) at the core of the super
resolution CNN. The architecture has been modified by adding layers near the beginning and end of the CNN that perform
normalization and dimensionalization of the data (respectively), enforce conservation laws, and ingest climatological data. The
core component of the CNN is the exact EDRN architecture however, which consists of two major components: a series of
"residual" blocks that build up a deep feature representation of the data (He et al., 2016) followed by an upsampling module that
increases the spatial resolution of the data based on the those features. We use the unmodified EDRN architecture at the core
of our CNN but out CNN construction means that other common super-resolution architectures can be substituted if desired,
or newer schemes can be used as they are developed.

The full super resolution CNN used here operates as follows: the initial input is passed through a normalization layer that
converts it to an approximately normal distribution followed by a single initial 9x9 convolutional layer with 64 output channels.
Note that this normalization layer applies a pre-defined operation depending on the data type and should not to be confused with
"batch-normalization." Then, the data are passed through a series of 16 residual blocks. These consist of a 3x3 convolution,
followed by a ReLU (rectified linear unit) transfer function, followed by another 3x3 convolutional layer, each with 64 output
channels. The residual blocks include skip connections that add the input tensor to their output. This creates a more direct path
for gradients to propagate through the CNN during training, which mitigates the vanishing gradient problem and allows very
deep architectures to train successfully. One additional skip connection is included that bypasses all of the residual blocks. The
residual blocks are followed by an upsampling module. The upsampling is done with two pixel shuffle operations, where the
number of channels are increased by a 3x3 convolutional layer and then the tensor is reshaped to convert the channel dimension
to larger spatial dimensions. A custom Keras layer was implemented to perform pixel shuffle with asymmetric increases along
the spatial dimensions. We found that the CNN performed best when the upsampling module was broken into a 4x5 pixel
shuffle, followed by a a convolutional layer, followed by a 2x2 pixel shuffle rather than a single 8x10 operation. Finally, the
tensor is passed through a 9x9 convolution with a single output channel, a dimensionalization layer, and then a custom layer
that enforces conservation rules between the CNN's high-resolution output and low-resolution input. The CNN is diagrammed
in Figure 1a, and the code can be found on the project's Github repository.[2]

The CNN is also used in configurations where high-resolution climatologically averaged mixing ratios are included as
additional inputs. When climatology is included, the data are merged with the sample data through channel concatenation both
at low resolution before the residual blocks and at high resolution after the upsampling module. Several convolutional layers
with ReLU transfer functions and max-pooling layers are used to process the climatology data. These are diagrammed in Figure
1b. The cosine of latitude is also included as an input in some cases, but is only used by the conservation law enforcement layer
meaning none of the layers with trainable parameters receive latitude data as an input (providing latitude data as an input to
CNN did not improve performance).

---

[2]https://github.com/avgeiss/chem_downscaling





We also demonstrate a VSR-CNN that has been modified for use with atmospheric data. These CNNs have similar internal structure to the SISR-CNNs but replace the 2D convolutions with 3D convolutions (Kim et al., 2019), use 256 channels within their residual blocks, and use only 12 residual blocks. Seven consecutive model time-steps are processed at a time with the primary goal of super-resolving the middle time-step. During training, the CNN is tasked to super-resolve all seven time-steps however, and their contributions to the loss are weighted such that the center time-step has the highest impact (the weights used

were: 1, 4, 16, 64, 16, 4, 1). At inference time, only the center time-step is retained as the output. CNN-based VSR schemes often include optical-flow-based motion vectors as inputs (Liu et al., 2020). Here, rather than computing motion vectors, we provide the 10m wind vector components and sea level pressure from the simulation as additional low resolution input channels. The wind vector components were standardized by scaling by a factor of 0.02 and the sea level pressure field is standardized by: $\widehat{SLP} = 2(SLP \times 10^{-5} - 1)$. Finally, a slightly different approach to including climatology was used for the VSR-CNNs. While

the SISR-CNNs were trained using randomly sampled spatial chips and including high resolution climatology as an input, the VSR-CNNs processes the entire global grid. This means that the CNNs can simply learn high-resolution climatologies from the training set using their layer biases. This an effective and simpler approach to including HR climatological data in the SR process, but is almost certainly more prone to over-fitting due to the training sample diversity that is lost without random sampling. No over-fitting was apparent here based on evaluation on the test and validation sets however. The VSR CNNs have

significantly larger memory requirements than the SISR-CNNs and were trained on a6000 GPUs while the SISR-CNNs were trained on RTX 2080ti GPUs.

## 3.2 Training on Log-Normally Distributed Data

Many trace chemical species have approximately log-normal mixing ratio distributions, meaning that across space and time their concentrations can span several orders of magnitude. This raises several issues when training a CNN, particularly when

trying to enforce conservation laws within the CNN. Neural networks often struggle to train on highly skewed or non-normal data distributions. There are two major factors: 1) CNNs train best when internal activations are approximately normally distributed. In particular, when outputs from convolutional layers are far from the non-linearity or deep in the saturated region of the following transfer function, gradients vanish, (Glorot and Bengio, 2010) which causes inefficient or failed training. Log-normally distributed inputs and outputs contribute to this problem. 2) when a conventional pixel-level loss function typically

used for super resolution, such as mean squared error, is applied to log-normally distributed output, the largest errors will correspond to the largest values in the output with orders of magnitude smaller contributions from grid-cells with lower values. In practice, the CNN prioritizes predicting the location of grid-cells with large values very accurately while mostly ignoring grid cells with smaller values, which is not usually desirable. A common solution is to train the CNN on data that has been standardized to a normal distribution and then re-dimensionalize the output from the CNN as a post-processing step. For the

chemical species considered in this study, a reasonable normalization procedure is to take the log of the dimensional data, then subtract the mean and divide by the standard deviation. A second problem then arises when enforcing conservation rules within the CNN however: conservation laws must be enforced on dimensionalized data. Ideally, we would like conservation laws to be enforced internally in the CNN so that the loss function can be applied to the final output and the CNN can learn to perform



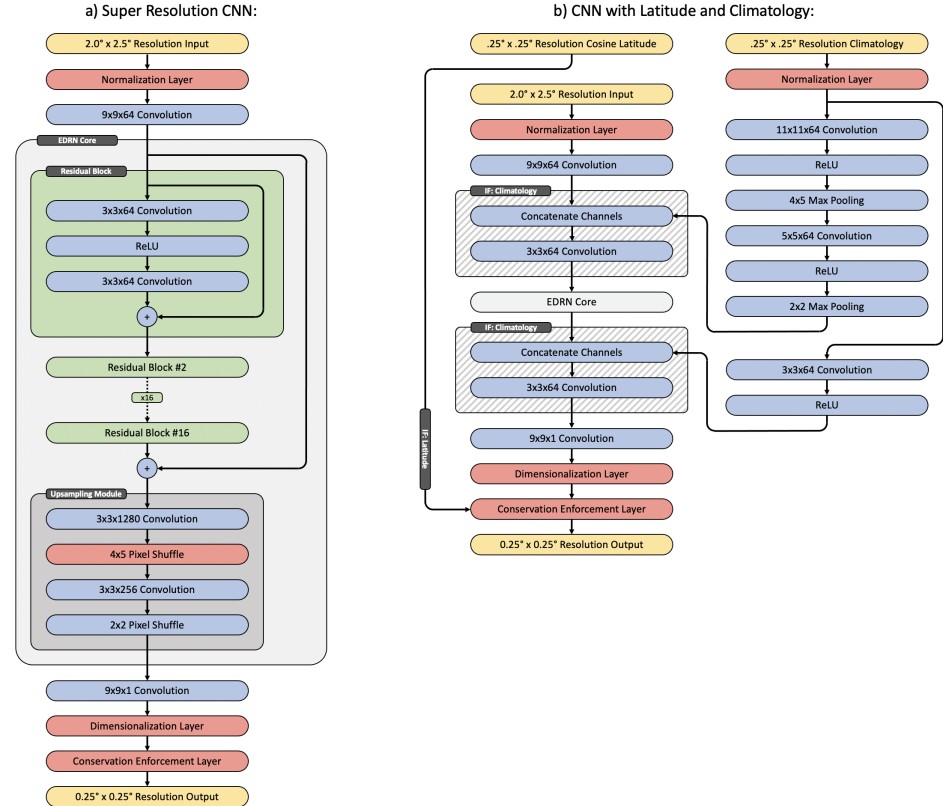

**Figure 1.** Diagram of the CNN architecture. Yellow cells denote input and output tensors, blue cells are commonly used Keras layers, red cells denote custom implemented layers, green cells represent residual blocks, and hashed cells denote layers that are conditionally on or off depending on whether climatological data are being used.

super resolution with this enforcement step. This is problematic if the CNN's internal representations and initial outputs are
standardized.

The approach used here for both learning and enforcing conservation rules on log-normally distributed data involves incorporating a normalization step and re-dimensionalization step into the CNN. The normalization layer performs this operation:

$$\hat{x} = \frac{1}{\sigma}\left(\log\left(x + \epsilon\right) - \mu\right) \tag{1}$$

here, the hat represents dimensionless data, mu and sigma are constants selected a-priori based on the mean and standard deviation of the natural log of each input variable (estimated using the training set). $\epsilon = 10^{-32}$, and is included to avoid taking the log of 0. The values of $\mu$ and $\sigma$ used in this study are given in Table A1. The output from the final convolutional layer in the CNN is not passed through a transfer function. Instead, a dimensionalization layer is used that applies the inverse of Equation





(1):

$$x = e^{\sigma \hat{x} + \mu} \qquad (2)$$

After the CNN outputs are dimensionalized they are passed to a layer that strictly enforces conservation rules and loss is computed on the output from that layer.

### 3.3 Training on $O_3$ Data

The $O_3$ data are not log-normally distributed. They are however, non-negative so a slightly different procedure is used to
super resolve the $O_3$ field. In this case, the normalization layer is removed and the $O_3$ data are simply scaled by a constant value of $4 \times 10^6$. This ensures that all values in the training set are within the range $[0, 1]$ (though most fall within $[0, 0.2]$). The dimensionalization layer is also removed and replaced by a sigmoid transfer function followed by division by the same constant. An alternative is to use the ReLU (rectified linear unit) or ELU (exponential linear unit) transfer functions which would also ensure non-negative outputs and would not cap the maximum value of the output, but we found that sigmoid works
better in practice. Other than these changes the CNN architecture and training procedure are unchanged for $O_3$.

### 3.4 Enforcing Conservation in the CNN

Here, we introduce the function used to enforce strict adherence to conservation rules when applying super resolution. This function is continuous and differentiable and is included in the CNN architecture during both training and inference. The approach is similar to Geiss and Hardin (2020b), however, the typical log-normal distribution of trace chemical species means
that this problem has slightly different constraints and a different operator is necessary. In particular, the outputs from the dimensionalization layer are bounded by $[0, \infty)$. While the mixing ratios technically should have an upper bound of 1, this does not need to be explicitly enforced for trace chemical species in practice. We use the following notation: $P$ is the mixing ratio in a single low resolution grid cell corresponding to an $N \times M$ region in the output, $x_i$ is the mixing ratio in a high resolution grid-cell output by the CNN within the $N \times M$ region corresponding to $P$, and $f(\mathbf{x}, P)$ is the high resolution output
after a continuous differentiable function $f$ has been applied to the initial output from the CNN. Here, $\mathbf{x}$ represents all $N \times M$ high-resolution output pixels corresponding to $P$. The function $f(\mathbf{x}, P)$ can be formulated:

$$f(\mathbf{x}, P)_i = x_i \left( \frac{P}{\overline{\mathbf{x}} + \epsilon} \right) \qquad \text{where:} \quad \overline{\mathbf{x}} = \frac{1}{NM} \sum_{x_j \in \mathbf{x}} x_j \quad \text{and:} \quad \epsilon = 10^{-32} \qquad (3)$$

This formulation of $f$ enforces the conservation rule:

$$P = \frac{1}{NM} \sum_{x_i \in \mathbf{x}} f(\mathbf{x}, P)_i \qquad (4)$$

This is accomplished by multiplying each block of output pixels by a constant $P/(\overline{\mathbf{x}} + \epsilon)$ that ensures the corrected output pixels sum to $P$. $\epsilon$ is added to the denominator to avoid divide by zero errors. This formulation of $f$ is differentiable with respect to $x$ which is crucial because this means it can be included in the neural network during training and gradients can be





back-propagated though this conservation enforcement layer to the trainable parameters within the CNN. $f$ also has the useful property that: $f(\mathbf{x}, P)_i \geq 0$, which prevents the CNN from generating non-physical negative mixing ratios.

## 3.5 Latitude Weighting

An additional concern when enforcing conservation laws on model output, is that the model uses a lat-lon grid. The grid cells do not have equal area, and cells near the equator will be significantly larger than cells near the poles. Ideally, this should be accounted for when taking a spatial average. For instance, when taking a 8x10 average over lat-lon grid cells the pole-ward cells should have a smaller contribution to the mean than the equator-ward cells because of their size difference. An additional term that weights the cell contributions to the mean can be added to Equations (3) and (4), such that Equation (3) becomes:

$$f(\mathbf{x}, \mathbf{\Phi}, P)_i = x_i P \, \bar{\mathbf{\Phi}} \left( \sum_{x_j, \phi_j \in \mathbf{x}, \mathbf{\Phi}} x_j \cos \phi_j + \epsilon \right)^{-1} \qquad \text{where:} \quad \bar{\mathbf{\Phi}} = \sum_{\phi_j \in \mathbf{\Phi}} \cos \phi_j \tag{5}$$

Here, $\mathbf{x}$ and $\mathbf{\Phi}$ represent the collection of grid point values ($x_i$) and latitudes ($\phi_i$) that correspond to the low resolution pixel/grid-point $P$. The summation in the definition of $f$ now takes a the average of the values in $\mathbf{x}$ weighted by the cosine of latitude. This formulation of $f$ now enforces a latitude weighted conservation rule:

$$P = \bar{\mathbf{\Phi}}^{-1} \sum_{x_i, \phi_i \in \mathbf{x}, \mathbf{\Phi}} f(\mathbf{x}, \mathbf{\Phi}, P)_i \cos \phi_i \tag{6}$$

## 3.6 Training Procedure

The data were divided into a training, test, and validation set. The simulation uses an hourly time-step and ran from 00:30z 01-Jan-2018 to 12:30z 16-Jun-2021. Data from 2018 and 2019 were used for training, 2021 used for validation, and 2020 used for testing. 2021 was used for validation because the run only included half of the year. This results in $17,520$ training samples, 8,784 test samples, and 3996 validation samples. Mean Absolute Error (MAE) computed on the validation set was monitored during training to ensure the CNNs were not over-fitting.

The SISR-CNNs were trained on 12,000,000 randomly selected samples using a mini-batch size of 12. This results in $1 \times 10^6$ total weight updates per training run. The Adam optimizer was used with an initial learning rate of 0.0001, $\beta_1 = 0.9$, $\beta_2 = 0.999$, and $\epsilon = 10^{-7}$. After 80% of the training the learning rate was manually reduced by a factor of 10. The CNNs use the mean squared error of the log of the outputs as a loss function:

$$\mathcal{L} = \overline{\left( \log\left( y + \epsilon \right) - \log\left( \hat{y} + \epsilon \right) \right)^2} \tag{7}$$

where $y$ and $\hat{y}$ are the ground truth and CNN predictions respectively and $\epsilon = 10^{-32}$ is included to prevent taking $\log 0$. We refer to this loss function as "LOG-MSE." The $O_3$ CNN simply uses MSE as the loss function.

The training procedure for the VSR-CNNs used identical validation split, optimizer, and loss functions, except that the $O_3$ VSR-CNN required $\frac{1}{2}$ the initial learning rate to ensure stability. A lower batch size of 4 was used due to GPU memory





limitations and the validation loss had stopped decreasing after only around 40,000 weight updates. The learning rate was reduced after the 32,000th training sample.

Training samples were generated for the SISR-CNNs by randomly selecting $256 \times 320$ size chips from the training set and reducing their resolution to $32 \times 32$ with 2-D averaging. No data augmentation schemes were used other than this random
selection of training chips. The CNNs were implemented in Keras to accept inputs of variable size, and the convolutional operations performed within the network are translationally invariant. This means that the CNN can be trained with smaller chips, instead of full global realizations, and then applied to the full model grid one time-step at a time during the testing phase. The VSR-CNNs were simply trained on global samples selected in random 7-hour chunks.

### 3.7 Evaluation

We compare the CNN downscaling to three common approaches for increasing data resolution. The first is simply nearest neighbor interpolation, which is included as an example of a worst-case downscaling approach. Secondly, we compare to bilinear and bicubic interpolators, which are frequently used for enhancing the resolution of images. Finally, we compare to an atmospheric chemistry downscaling scheme that is capable of incorporating high-resolution chemical climatologies (Geddes et al., 2016) (referred to as "Clim." below). The approach projects the coarsely resolved modeled mixing ratios, which capture
the transient changes in concentration, onto the finely resolved spatial pattern of climatological mixing ratios. For chemical species like $NO_2$, that have a very persistent spatial distribution with strong gradients, this approach significantly outperforms interpolation. The version of the climatological average scaling (e.g. Geddes et al. (2016)) scheme used here does not include the smoothing operator, which increases the visual quality of the result. By omitting this smoothing step however, this climatology-based downscaling approach gains the same conservation properties as the CNNs used here: the low resolution
input data will be exactly reproduced under 2D averaging.

The downscaling CNN was evaluated using grid-cell-wise mean absolute error (MAE) of the outputs. Exact reconstruction of the high-resolution data is impossible, and even outputs with very low MAE may appear spatially smoothed, so we also evaluated results using two other metrics that approximate how realistic the spatial structure in the downscaled output is. The first is the structural similarity index (SSIM). SSIM approximates the visually perceived difference in spatial structure between
two images (Wang et al., 2004). It is a dimensionless value and scales between -1 and 1, with an SSIM of 1 representing an exact match between the two images. The SSIM is constructed from three components measuring differences in luminance, contrast, and structure (in the context of images) evaluated using a moving window, and then spatially averaged. Because the mixing ratio data used here is log-normally distributed we compute the SSIM on the log of the concentrations ("LOG-SSIM"), otherwise there is only limited perceptible spatial structure (except in the case of $O_3$).

Finally, zonal power spectral density (PSD) of the downscaled data, averaged meridionally, was evaluated. The PSD is defined here as:

$$\text{PSD} = 10\log_{10}\left(\overline{|\mathcal{F}_\lambda\left\{\log\hat{y}\right\}|^2}\right) \tag{8}$$





where $\log \hat{y}$ represents the natural logarithm of the high-resolution CNN output, $\mathcal{F}_{\lambda}$ represents the Fourier transform taken with respect to longitude, and the overbar represent averaging with respect to latitude and over all test samples. The PSD provides

an estimate of the spatial variability and sharpness of features recovered by the downscaling schemes. A PSD curve that closely approximates the PSD curve of the ground truth data implies a higher-fidelity result. The performance of the CNNs and various benchmark schemes for MAE and LOG-SSIM are shown in Table 1 and Figure 5 and the PSD curves for the various schemes are shown in Figure 6.

## 4 Experiments

The super resolution CNNs were trained separately for each compound studied and were train in several different configurations to assess the impact of enforcing conservation rules, latitude-area weighting, and including climatology as an input. These different variations on the SISR and VSR CNNs are denoted with the labels:

**"Ctrl"**: Control experiments that use the mostly unmodified EDRN (Lim et al., 2017) architecture.

**"Enf":** A layer is added to the end of the CNN to enforce conservation rules using equation 3.

**"Lat"**: Like "Enf" but equation 5 is used instead to account for non-uniform grid-cell areas.

**"Cli"**: High resolution climatologies are provided as an input to the CNN.

For the SISR-CNNs we examine five cases. A control run with the typical EDRN architecture is compared to a training run with conservation law enforcement to determine the impact of the enforcement layer. Another CNN is trained with latitude weighted conservation law enforcement to determine the significance of latitude weighting. Finally, two SISR-CNNs are trained with

climatology data both with and without the conservation enforcement layer. Only two experiments were performed with the VSR-CNNs, one with and one without conservation law enforcement. Recall however, that the VSR-CNNs are constructed in such a way that they can memorize the training set climatology. Errors computed on the test set for each of these CNN configurations and each of the benchmark schemes are shown in Table 1.

## 5 Results

Overall, the deep-learning schemes significantly outperform the conventional downscaling and interpolation methods (Table 1). Figures 2 and 3 show sample outputs from the best performing super resolution scheme for each compound (typically the $VSR_{Ctrl}$ scheme) alongside coarsely resolved and ground truth mixing ratio data. Example outputs from every downscaling scheme for an $NO_2$ sample case are shown in Figure 4. Each of the sample cases shown were chosen manually from the

test set and were selected to include features of interest that are difficult to super-resolve such as high-concentration plumes





| Mean Absolute Error | | | | | | | | | | | |
|---|---|---|---|---|---|---|---|---|---|---|---|
| | Interpolation/Downscaling | | | | SISR-CNNs | | | | | VSR-CNNs | |
| | Nearest | Bilinear | Bicubic | Clim. | Ctrl | Enf | Lat | Ctrl/Cli | Lat/Cli | Ctrl | Enf |
| $NO_2(ppbv)$ | 0.150 | 0.144 | 0.145 | 0.103 | 0.090 | 0.092 | 0.092 | 0.079 | 0.081 | **0.057** | 0.058 |
| $SO_2(ppbv)$ | 0.186 | 0.177 | 0.186 | 0.136 | 0.118 | 0.123 | 0.123 | 0.104 | 0.108 | **0.073** | 0.074 |
| $CO(ppbv)$ | 4.323 | 4.350 | 4.186 | 3.858 | 2.811 | 2.842 | 2.847 | 2.709 | 2.733 | **2.296** | 2.299 |
| $O_3(ppbv)$ | 1.232 | 1.291 | 1.223 | 1.196 | 0.900 | 0.883 | 0.885 | 0.879 | 0.850 | 0.766 | **0.701** |
| $PM_{2.5}(\mu g m^{-3})$ | 1.548 | 1.518 | 1.450 | 1.451 | 1.031 | 1.047 | 1.047 | 1.008 | 1.024 | **0.855** | 0.863 |
| LOG-SSIM | | | | | | | | | | | |
| $NO_2$ | .772 | .773 | .785 | .862 | .902 | .903 | .903 | .929 | .930 | **.959** | **.959** |
| $SO_2$ | .834 | .837 | .843 | .863 | .911 | .911 | .912 | .922 | .922 | **.945** | **.945** |
| $CO$ | .935 | .939 | .945 | .949 | .971 | .971 | .971 | .973 | .973 | **.980** | **.980** |
| $O_3$ | .865 | .872 | .884 | .877 | .920 | .922 | .922 | .926 | .930 | .941 | **.951** |
| $PM_{2.5}$ | .832 | .841 | .853 | .838 | .917 | .918 | .918 | .921 | .921 | **.947** | **.947** |

**Table 1.** Performance of the downscaling CNNs compared to interpolation and conventional downscaling. The top section shows pixel level mean absolute error (lower values are better) and the bottom section shows the structural similarity index computed after taking the log of the log-normally distributed variables and scaling the data to a 0-1 range (higher values are better).

downstream from urban areas and particularly sharp gradients due to weather features like strong cold fronts. While the sample cases were not chosen randomly they were all chosen prior to evaluation of any of the downscaling schemes.

The SISR-CNN schemes that incorporated high-resolution climatological data performed significantly better than any other SISR scheme in terms of both MAE and LOG-SSIM. The MAEs of the climatology driven CNNs were 54% ($NO_2$), 56% ($SO_2$), 65% ($CO$), 70% ($PM_{2.5}$), and 70% ($O_3$) of the MAE of bicubic interpolation. The reason for this is clear in Figure 4, which shows output from each of the SR-schemes for a single test case of $NO_2$ mixing ratios. The test case shows $NO_2$ concentrations over South America, and the South Atlantic and a portion of the Southern Ocean (the precise times and locations for each of the test cases can be found in Table A2). While all of the CNN-based schemes (Figure 4 panels a, b, d, e, f, h, and i) can reconstruct the larger features in the sample with much higher fidelity than the interpolation schemes (panels k and l), the climatology-driven CNNs (Figure 4 panels a, b, e, and f) are able to incorporate very small scale features that can only be determined from the high resolution climatology. In particular, the point sources associated with small cities, islands, and ship tracks are incorporated into their output while these features are blurred by most of the other schemes. The climatology-based downscaling scheme is able to reproduce many of the stationary small-scale features as well, but cannot sharply resolve transient features associated with atmospheric motions (Figure 4 panel g). Essentially, it represents the very small-scale stationary features at the target resolution but simultaneously represents the large-scale transient features at the resolution of the input data. The other CNN-based schemes (Figure 4 panels d, h, and i) are not able to reconstruct these very

**Figure 2.** Select sample cases from the test set. Each row represents a different compound while the left column shows the coarsened data, the middle column shows the super resolved output from the best performing CNN for that compound, and the right row shows the ground truth.

small features, but they do produce much sharper downscaled data than interpolation. They are particularly good at localizing sharp gradients that span multiple pixels: the large, distinct, plumes in the southern ocean and ship tracks extending from Southern Africa towards the North and Equatorial Pacific are are both good examples of this.

The VSR schemes provide yet another significant performance advantage over the best SISR-CNNs. They improve the pixel-MAE to 40%($NO_2$), 39% ($SO_2$), 53% ($CO$), 56% ($PM_{2.5}$), and 54% ($O_3$) that of bicubic interpolation. While the SISR-CNNs with climatology input are able to accurately resolve small-scale sources of each of the chemical species the VSR-CNNs are able to leverage the time-evolution of the low-resolution data to infer the locations of small-scale plumes emanating from





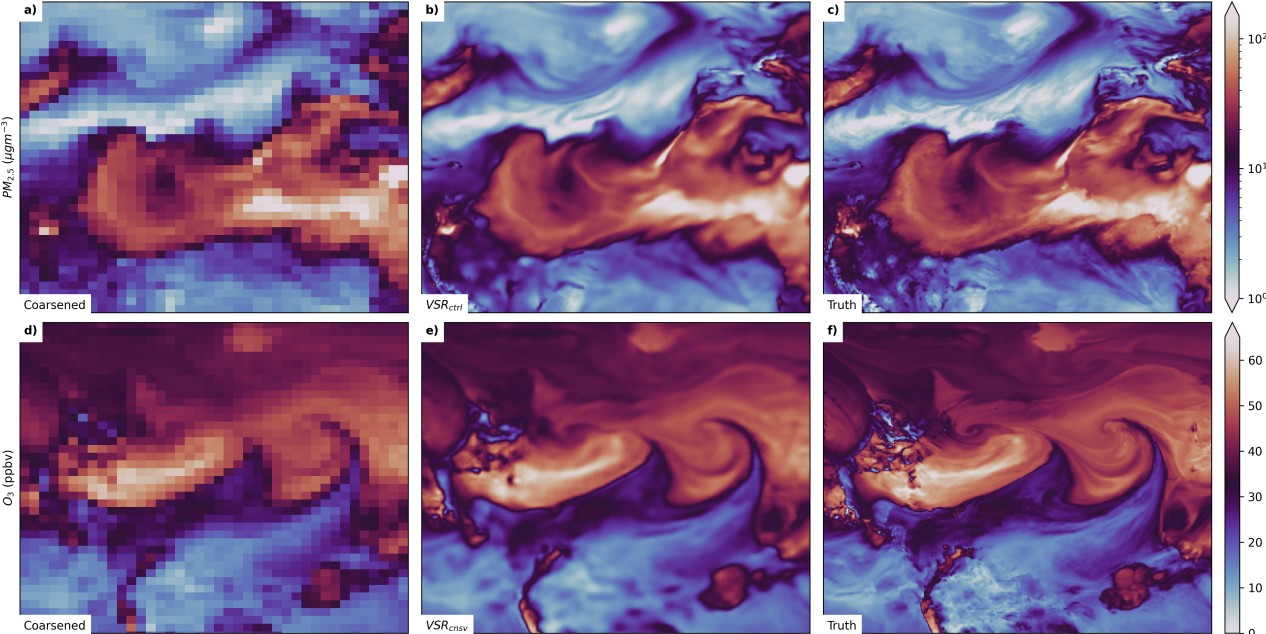

**Figure 3.** sample cases for $PM_{2.5}$ and $O_3$. Same as Figure 2.

these sources. They are also able to much more accurately resolve small-scale gradients in the transient weather features in the
simulation.

Strict enforcement of conservation rules did not lead to improvement in performance for most SISR-CNNs. The CNNs that
included climatology performed from 1.0% to 3.7% worse in terms of MAE. This was also the case for the CNNs that did not
include climatology as an input. Despite this slight reduction in MAE, the SSIM was not substantially altered by enforcing
conservation rules. In the case of $O_3$, which was not log-normally distributed, the MAE and SSIM were both improved by
enforcing conservation laws (MAE by 3.3 %). This slight improvement in skill is more consistent with the results of Geiss and
Hardin (2020b). This discrepancy is likely related to the log-normal distribution of most of the mixing ratios. Conservation
laws are enforced on the dimensional data (and not the standardized data processed by the CNN), and the high resolution
dimensional samples are dominated by a handful of grid-cells with very high concentrations while variability between other
grid-cells is minimal. Enforcing conservation laws on this type of data means that cases with extremely high mixing ratios in
the original HR data will tend have these high concentrations spread across all the pixels corresponding to the LR grid cell in
the input to some degree. Even so, the CNNs seem to have mostly learned to account for this because the increase in MAE
is very small and while there may be some indication of the location of the LR grid cells for the conservation law enforcing
CNNs in Figure 4, any artifacting from this effect is nearly imperceptible. Meanwhile, the VSR schemes did not show such
a pronounced difference between the cases with strict enforcement of conservation rules and cases without. While the "Ctrl"
cases did perform better the change in MAE was only a fraction of that for the SISR-CNNs and there was no change in SSIM.



**Figure 4.** Super resolved $NO_2$ concentrations for 11:30Z July $20^{th}$ 2020 in the simulation from 70.25-4.75S and 75-225E.





This is encourgaing, and implies that improving the overall accuracy of the super-resolution scheme reduces any negative impacts from enforcing conservation rules.

Using latitude weighting when enforcing conservation rules had no substantial effect on the CNNs' skill. This is unsurprising because the latitude weighting does not dramatically change contributions for neighboring grid cells when enforcing
conservation rules. In most locations (except very close to the poles) the grid cell areas are nearly constant over a 2-degree change in latitude.

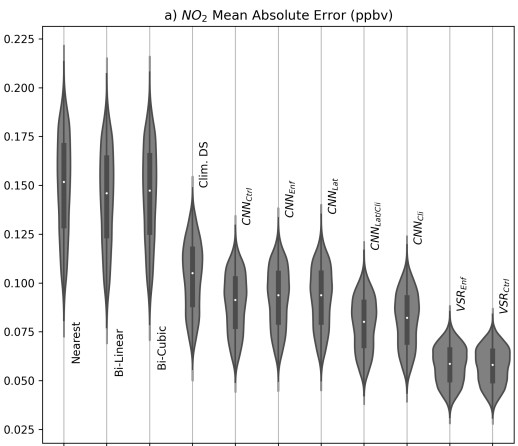

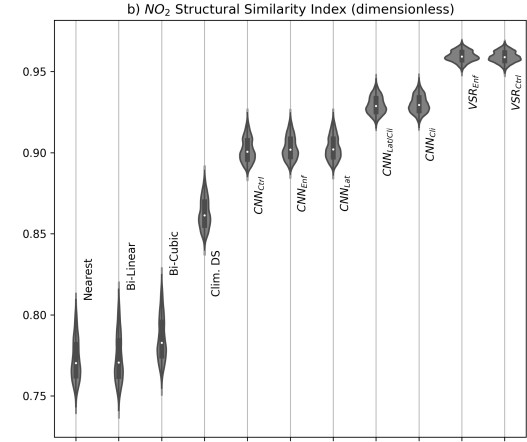

**Figure 5.** Distribution of mean absolute error and SSIM with respect to individual test cases for $NO_2$

In addition to showing the mean MAE and LOG-SSIM in Table 1 we show the distribution of these metrics across the samples in the test set using violin plots in Figure 5. This plot demonstrates the variability in skill due to individual samples. There is relatively high variability in MAE and the best-case samples for the interpolation schemes have lower MAE than the
worst-case samples for the CNNs. Note that there are no individual cases where bi-cubic interpolation outperforms the best SISR-CNN however. There is significantly less variability due to sample variance in SSIM, and the distribution of SISR- and VSR-CNN SSIMs do not overlap the SSIM distributions from the interpolation schemes at all.

While high pixel-level accuracy is desirable for downscaling schemes, perfect pixel level accuracy is not achievable. The process of reducing the resolution of data irreparably destroys information, and while some small-scale features can be inferred
by super resolution schemes, at least a portion of the high-resolution information will not be recoverable. In addition to evaluating pixel-level accuracy we analyze the power spectral density (PSD) of the downscaled outputs. PSD indicates the distribution of energy in frequency space of the outputs. If the SR scheme's outputs have a very similar PSD curve to the high resolution data this implies that while it may not be correct at a pixel level, the SR data has a realistic distribution of variability across spatial scales. For some atmospheric processes and downscaling applications, it may be a priority to adequately represent small
scale features, like turbulent motions for instance, even if they are not generated at exactly the correct location.





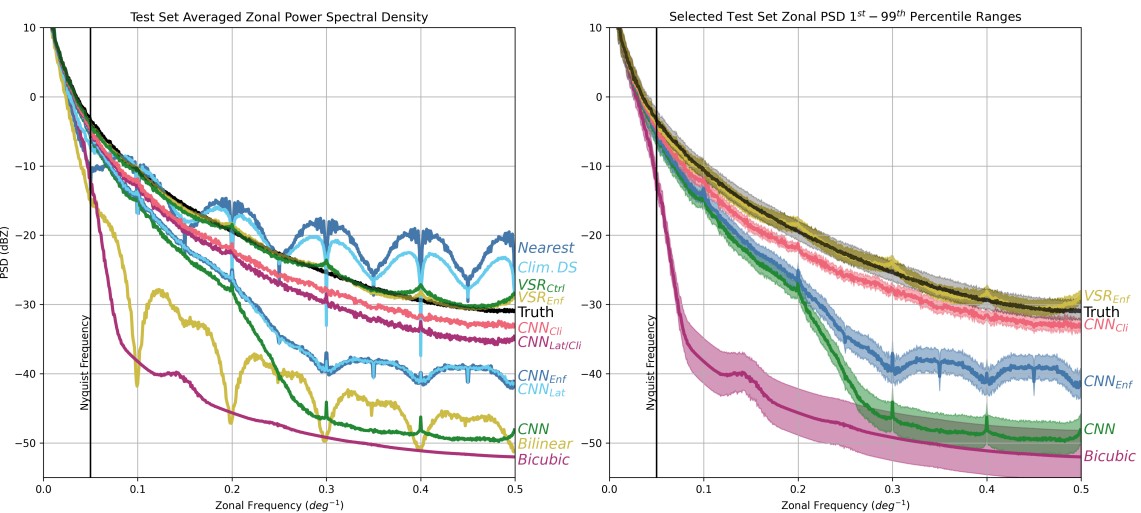

**Figure 6.** Panel a: zonal power spectral density (PSD) curves (averaged meridionally) for outputs from the various downscaling schemes. The black curve shows the PSD of the ground truth data and close proximity to the black curve indicates that a scheme successfully reproduces the spatial variability of the original data across multiple scales. The vertical black line indicates the highest frequency that can be represented by the coarsened input data. The CNNs that ingest high resolution climatology data and the VSR-CNNs perform the best. Several of the schemes show ringing artifacts due sharp discontinuities in the output associated with the coarsened grid of the input data. Panel b) the shaded region indicates the 1-99th percentile range for PSD curves computed for all samples in the test set, showing the dependence of the result on sample variability.

Figure 6a shows the PSD curves for the high resolution simulation of $NO_2$ (black) along with curves for the various super-resolution and conventional downscaling schemes. An ideal downscaling scheme's PSD curve will exactly match the ground truth curve. Of all the CNNs the two VSR CNNs perform the best, very closely matching the ground truth PSD curve. The two CNNs that were provided with high-resolution climatology as an additional input also perform significantly better than any other schemes. They both have slightly lower energy at higher frequencies than the ground truth, indicating that while they produce very realistic variability they are not able to capture all of the small-scale variability in the original data. Additionally, these two curves vary smoothly with respect to frequency and do not show spectral artifacts seen for several of the other downscaling schemes. This indicates that they are resistant to artifacting due to the scale of the low resolution data.

The bilinear and bicubic interpolation schemes heavily smooth their outputs, and have significantly lower power at high frequencies than any of the others. Simultaneously, the other two worst-performing schemes, nearest neighbor and climatological downscaling, have higher power at these high frequencies but in this case it is due to spectral artifacts. Both of these schemes heavily pixelate their outputs and the sudden jumps in concentration in their outputs require high frequency components to





represent in Fourier space. While they have high energy at high frequencies, it is not because they are accurately reconstructing the true high-frequency variability in the ground truth.

The control CNN experiment has a better PSD curve than the interpolation schemes up until about 4 times the Nyquist frequency after which it rapidly drops off, whereas the PSD of the interpolation schemes rapidly decrease beyond the Nyquist frequency. The CNNs with conservation law enforcement have slightly higher PSD at high frequencies than the one without, but show evidence of artifacts similar to bilinear interpolation. Because the conservation law enforcement operates on 8x10 pixel blocks, some evidence of the location of these blocks is noticeable in the high resolution output, and this leads to the
artifacting. The more skilled CNNs that enforce conservation (CNN-Enf/Clim and VSR-Enf) do not show much evidence of this however, which seems to indicate that the increased skill and ability to reproduce small-scale features reduces the likelihood of pixelation-like artifacts in the output. Both the SISR-CNN that ingest climatology data and VSR-CNNs very closely match the PSD curve for the ground truth samples. The VSR schemes are the closest but show a weak periodic signal due to slight pixelation in their output. In any case, all four of these CNNs very realistically reproduce the zonal variability in the training
dataset even at high frequencies.

Figure 6b shows PSD curves from several select experiments with a 1st-99th percentile shaded region around the curve computed over all the samples in the test set. This panel indicates that there is relatively little variability in these curves dues to sample variability in the test set. The shaded regions very tightly follow the mean, particularly for the higher accuracy CNNs.

Lastly, the performance of the VSR schemes is particularly notable. They not only provide a significant performance en-
hancement over the SISR CNNs in terms of the quantitative metrics presented in this section, but also produce visually striking results (Figures 2 and 3). Some notable features are the cyclone in the upper right quadrant of Figure 2 panel b, the plumes emanating from small point sources in the upper portion of Figure 2 panel e, and the sharp gradients associated with South Atlantic ship tracks and Southern Ocean plumes in Figure 2 panel h. Many of these features seem near-impossible to infer from the low resolution inputs shown to the left of each of these panels, but by incorporating time dependence, climatology, and
model state variables the CNNs are able to do it. We have provided a video supplement[3] that animates $O_3$ output from several of these super resolution schemes, and the VSR scheme in particular is able to produce high resolution results with smooth time continuity, and closely emulates the high-resolution simulation. To the best of our knowledge, VSR CNNs have not yet been used to downscale atmospheric data, and their success here, and superiority to SISR methods, indicates that they will be an exciting area of research moving forward.

## 410   6   Discussion and Conclusions

CNN-based super resolution schemes can very accurately downscale atmospheric chemistry simulations. In this work, we demonstrated several new important developments: CNN-based super resolution schemes can be effective for downscaling for large resolution changes. Most of the CNN-SISR literature focuses on relatively small changes (2x to 4x) and here we have shown that the same schemes can be applied to perform a 8x by 10x resolution increase to downscale data between two

---

[3]https://youtu.be/JPJX1k-5yew





common GCM grid resolutions (2.0x2.5 degrees and 0.25x0.25 degrees). This also demonstrated that asymmetric resolution changes are feasible simply by modifying the pixel-shuffle upsampling method. We also demonstrated a new method for strictly enforcing adherence to physical conservation laws in downscaling CNN outputs. Developing such techniques will be crucial for applying the capabilities of modern machine learning schemes to physical science data. Implicitly enforcing adherence to known physical laws within ML architectures can enhance the trustworthiness of these schemes, and in some cases, may

improve their accuracy. We incorporated normalization and dimensionalization layers into our CNN architecture which allowed it to perform enforcement of conservation laws on log-normally distributed data, and also found that in the case of $O_3$ data (that is not log-normally distributed) enforcing conservation rules can actually improve the performance of the super resolution schemes. Finally, our results demonstrate that incorporating the time-evolution of the data into the super resolution scheme, in this case by using a 3D-convolution based VSR-CNN, can provide a significant improvement in performance over SISR

schemes. Most past research that has used CNNs for downscaling has focused on SISR schemes and this work shows that VSR-based super resolution should likely be the focus moving forward. 3D-convolutions are a natural choice for representing the spatial and temporal dependencies in the data, and data produced by atmospheric simulations is also a good candidate for VSR-CNN methods because the wind vectors and potentially other atmospheric state variables can be provided as additional inputs to the CNN.

While CNNs represent a huge leap forward in the accuracy of downscaling algorithms, they have several drawbacks that should be addressed. The first, that is specific to our approach, is that some of the CNNs trained here, particularly those that enforce conservation rules, have a tendency to introduce artifacts in their outputs at the scale of the grid used to generate the inputs. We did notice a promising result that the magnitude of this artifacting was reduced for the more accurate downscaling CNNs (the VSR-CNN and SISR-CNN$_{Clim}$ experiments), but it was not completely removed. In future applications, a potential

solution is to modify the loss function with a regularizer term that penalizes larger than average spatial gradients every Nth pixel (where N is the downscaling factor), though this may lead to slightly reduced performance. There are also several general limitations of using CNNs for downscaling. One is unpredictable behavior on out-of-sample data, or the "covariate shift" problem (McGovern et al., 2021). This could be a particularly big issue if a model is trained on a simulation of the current climate and applied to simulations in future climate, where the climatological state of the atmosphere may have changed.

This should be carefully considered when using the VSR method demonstrated here because it is designed to memorize the high-resolution climatology of the training data. For atmospheric chemistry simulations specifically, long term changes in point sources due to human activity are another long-term shift that would need to be addressed. Potential solutions to these problems include: providing climatology as an input and recomputing it depending on the time-period being studied, training the CNN on a much longer simulation that includes different climate states, or using data augmentation to change the magnitude of certain

input fields during training (though this would require making predictions about how those input fields would be modified in a future climate). We note that the downsampling enforcement constraint used in this study does at least partially mitigate this problem, because while CNNs may lose skill on out-of-sample data they mathematically cannot produce outputs that do not downsample to the original input data. Finally, there are differences between the data produced by low-resolution simulations and by downsampling high-resolution simulations. At the very least, the distributions of the input and training data should





be analyzed before using a downscaling CNN in this way. Two potential solutions to this issue are training using simulations performed across multiple resolutions (using a nested mesh for example) or using a conditional generative adversarial network (CGAN) to focus the CNN on producing plausible small scale variability with less dependence on pixel-level errors (Wang et al., 2021).

     While the downscaling shown here is a dramatic improvement in skill over both conventional downscaling schemes and
SISR-CNNs, there is room for further improvement. The skill of the VSR CNNs can almost certainly be further improved through additional super-parameter tuning and further experimentation with the CNN architecture. While there is a large body of research in VSR schemes virtually none has been done on applying VSR-CNNs to atmospheric simulations. GCM data have fundamental differences from video data, for example: atmospheric motion vectors and other state variables can be provided directly to the SR schemes instead of motion vectors estimated from optical flow, atmospheric data do not require any
frame alignment step, the governing equations underlying atmospheric motions are known, and high-resolution climatology is known. Developing VSR schemes designed and tuned specifically for use with atmospheric data is a promising path forward. Additionally, other existing CNN techniques may produce improved results. For example, many SISR-CNNs use adversarial loss functions and this may further increase the fidelity of the output from atmospheric downscaling schemes. The method for enforcing physical constraints on the outputs that was introduced here does not preclude the use of an adversarial loss
function, meaning there is potential to develop VSR CNNs that hallucinate hyper-realistic small-scale variability while strictly adhering to the output of the physics-driven low-resolution simulation. This would be a significant step towards improving the trustworthiness of GAN-based downscaling, which has been identified as a key issue when applying GANs to scientific data (McGovern et al., 2021).

     In closing, CNNs and in particular VSR CNNs represent a leap forward in the capabilities of atmospheric downscaling
methods. The approach shown here could dramatically reduce the computational and energy cost of running simulations at very high resolution. Near-surface concentrations of the compounds studied here have a significant impact on human health and accurately resolving high resolution features, like plumes, is crucial for air-quality forecasting. Furthermore, global chemistry simulations play a key role in understanding future weather and climate, and accurately resolving fine-scale features is critical to this effort. Here we have demonstrated a method for strictly enforcing conservation law constraints in a CNN architecture
without significant loss of accuracy. While current super-resolution CNNs can produce aesthetically pleasing high-resolution outputs, development of CNN architectures targeted towards physical science problems and capable strictly enforcing physical constraints will be essential for developing deep-learning methods compatible with earth-science problems, and capable of generating trustworthy and actionable predictions. Using the methods introduced here, we envision producing high-accuracy and high-resolution atmospheric chemistry simulations and ensemble forecasts at a fraction of the current cost.



*Note: I will generate DOIs and permanent links to the code, trained models and video supplement after review in case these items change as a result of reviewer feedback. -A. Geiss*

*Code availability.* The code used for this project is available from:

`https://github.com/avgeiss/chem_downscaling.`

Definitions of custom Keras layers can be found in the file: `neural_networks.py`

*Data availability.* The NASA GEOS-CF data used in this work is available at:

`https://portal.nccs.nasa.gov/datashare/gmao/geos-cf/v1/das/.`

Trained CNNs can be downloaded from:

`https://drive.google.com/file/d/17xqrUFdXo0uarEXLTuYA4E1eOdF2EEy8/view?usp=sharing.`

Note that for training the SISR CNNs ingest HR data and downsample it themselves before super resolving it, so this layer will need to be
removed before application to coarse data.

*Video supplement.* An animation of simulated, coarsened, interpolated, and super-resolved $O_3$ mixing ratios is available from:

`https://youtu.be/JPJX1k-5yew`

*Author contributions.* AG performed experiments, developed method, wrote manuscript. SJS conceived project, wrote and edited manuscript, acquired funding. JCH contributed to experimental design, edited manuscript

*Competing interests.* The authors declare no conflict of interest.

*Acknowledgements.* A portion of the research described in this manuscript was conducted under the Laboratory Directed Research and Development Program at Pacific Northwest National Laboratory (PNNL), a multiprogram national laboratory operated by Battelle for the U.S. Department of Energy. SJS is grateful for the support of the Linus Pauling Distinguished Postdoctoral Fellowship program. The Pacific Northwest National Laboratory is operated for the U.S. Department of Energy by Battelle Memorial Institute under contract DE-AC05-
76RL01830.





# Appendix A: Additional Tables

|   | $NO_2$ | $SO_2$ | $CO$ | $PM_{2.5}$ |
|---|--------|--------|------|------------|
| $\mu$ | -25.1 | -24.4 | 16.4 | 1.0 |
| $\sigma$ | 3.0 | 7.0 | 0.3 | 2.0 |

**Table A1.** Normalization constants for log-normally distributed chemical species.

| Sample Cases | | | | |
|---|---|---|---|---|
| Compound | Shown In: | Date | Latitude Range | Longitude Range |
| $CO$ | Fig. 2(a-c) | 11:30Z, $1^{st}$ Jan. 2020 | 4.0°S-76.0°N | 75.0°E-175.0°E |
| $SO_2$ | Fig. 2(d-f) | 11:30Z, $4^{th}$ Mar. 2020 | 14.0°S-66.0°N | 259.75°E-359.75°E |
| $NO_2$ | Fig. 2(g-i) & 4 | 11:30Z, $22^{nd}$ Aug. 2020 | 79.0°S-1.0°N | 100.0°E-200.0°E |
| $PM_{2.5}$ | Fig. 3(a-c) | 11:30Z, $18^{th}$ Feb. 2020 | 1.5°S-78.5°N | 37.5°W-62.5°E |
| $O_3$ | Fig. 3(d-f) | 11:30Z, $4^{th}$ Apr. 2020 | 11.5°S-68.5°N | 75.0°E-175.0°E |

**Table A2.** Dates and lat-lon boundaries for the sample cases shown in Figures 2, 3, and 4.



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
