# Peer review of "Downscaling Atmospheric Chemistry Simulations with Physically Consistent Deep Learning"

_Geoscientific Model Development, 2022_

## Author Comment (AC4)

**Comment to reviewers/editor:**

We would like to sincerely thank the reviewers and the editor for taking the time to review our work and provide thoughtful and helpful feedback. Both reviewers suggested useful edits to the manuscript and we have integrated all of them and uploaded both a revised manuscript and a tracked changes document. We have provided line-by-line individual responses to each reviewer below.

-Andrew Geiss, Sam J. Silva, and Joseph C. Hardin

**Specific Responses to Reviewer 1:**

*Section 1.1 and 3.1: I think the introduction of the technical language could be improved a bit. GMD will have readers who don't understand what CNNs are or why we'd want to use one for image processing. I think perhaps a couple of plain text sentences about CNNs and why they're useful would benefit this section. There's also language used without introduction such as 3-layer, vanishing gradients, convolutional kernels, and deeper CNNs, that could be explained more. I don't disagree at all with what you've written, I just think the audience (GMD) and the manuscript will benefit from a little more explanation.*

This is a good point, and a common oversight in these types of interdisciplinary papers. There is some basic background on CNNs in the introduction on lines 43-44 and we have added some exposition whenever technical machine learning terminology is introduced in the revised manuscript. This is probably easiest to see in the tracked changes document rather than listing line numbers here.

*Minor comments*

*L92: It's not immediately apparent to me why being able to train a CNN on log-normally distributed data is a result of your work. Wouldn't a standard approach be to scale your data before training?*

Yes, it is common practice to scale inputs and then re-scale outputs after processing. Here we simply include these scaling/re-scaling steps as components of the CNN so they do not need to be done on CPU before and after applying the CNN. It is not a particularly major contribution, but it is a departure from the norm. Building CNNs this way could make them much easier to apply for an end user because they can be applied directly to dimensional model output fields and will return a similar field with the correct units. Normalizing this type of design choice could help smooth integration of CNN-based algorithms into applications in the environmental sciences and make things easier for users without a strong background in machine learning. Finally, in our specific case, re-dimensionalizing the initial output from the CNN is necessary before passing it to the conservation enforcement layer. We have added some comments about this on lines 208-210 and changed the discussion on line 100 to clarify this.

*L160: What's a spatial chip?*

A randomly selected 32x32 grid-cell patch of data (or 256x320 at high-resolution). We have added a better definition to the manuscript on line 170.

*L174&177: There are numerous other loss functions that will account for the issue of the loss dominating for large concentrations (negative log likelihoods, normalised loss etc)*

Good point, we have added a comment about this on line 188

*L201: Unit for this value 4x10^6 would be useful?*

This is dimensionless, the O3 data are number concentrations. Added a note of this on line 214.

*L203: Neither a ReLU nor an ELU will enforce non-negative outputs. I believe this sentence to be incorrect and it should be updated.*

ReLU produces outputs >= 0. ELU >= -1. So actually, ELU +1 is needed to enforce non-negativity. This has been corrected.

*L271: What was the motivation for using MAE to evaluate. I personally would have thought MSE alongside so evaluation of fractional errors to be more informative.*

We opted to use MAE to evaluate so that errors can be reported in physical units of number concentration. Also, MSE or RMSE will mostly represent the error in the handful of pixels with high concentrations because the concentrations of these compounds can span multiple orders of magnitude. We have added a comment to this effect on lines 289-291.

*L275: This could be a limitation in my understanding, but I thought SSIM values are between 0 and 1, not -1 and 1.*

An SSIM of 1 occurs with perfectly correlated images and 0 occurs for uncorrelated images. The covariance term in the SSIM formula can lead to negative values when two images are anticorrelated however. This is not particularly relevant to our work, but negative SSIM is possible. A note of this is added on line 295.

*Table1: What's the intuition behind CO and it's fairly similar performance (for LOG-SSIM, but not MAE) across the downscaling methods?*

CO appears to be an easier field to super resolve, perhaps because of less complicated spatial distributions than say NO2. The control CNN has an SSIM of .971 and SSIM is bounded by 1 so it can't improve much more. VSR improves on SSIM by 0.009 or 31% of the possible improvement. Meanwhile MAE is bounded by 0 and VSR improves on the control CNN by 18%. In any case, SSIM is a very different metric than MAE and while I would expect it to improve when MAE improves, I would not expect this relation to be linear.

*Figure 2 and others: Representing ship tracks only really shows that the CNN learns that these are stationary features right. If we moved this ship track elsewhere, I'd imagine the CNN wouldn't upscale that well. Is this right?*

Interesting question. We used several different treatments of how the CNNs learn about climatology in this study. The VSR CNN trains on global samples so it can memorize climatology. Some of the SISR CNNs take high resolution climatology as an input. In both of those cases it's probable the CNN is simply memorizing ship track locations (which isn't necessarily bad since these are static in the model and shipping lanes remain relatively static in the real world). The SISR CNNs without a climatology input (see Figure 4 panels d, h, and i) are trained of randomly selected spatial samples from the training dataset and to these CNNs the ship tracks are not stationary features. These CNNs still do a very good job of localizing the sharp lines associated with the ship tracks. If you refer to panels a, b, e, and f (CNNs that can memorize climatology), the high-res outputs from these CNNs go a step beyond ship tracks and include individual point sources in the ship tracks that are stationary in the model. In summary, it looks like the ship track locations can be recovered by the CNNs without memorizing climatology, but the grid-cell-scale point sources only show up when the CNN has access to high-resolution climatology. If you look at some of the examples from CNN SISR research on images (Zhang 2018 https://arxiv.org/pdf/1802.08797.pdf – has some nice examples on the Urban100 dataset),

these algorithms are generally very good at sharply resolving linear features that span multiple coarse pixels, and as long as the ship track is at least visible in the coarse data it is reasonable to think that the SR-CNNs could resolve it without relying on memorization. We have added some discussion of this point on lines 350-361.

*Figure 6: About the 'ringing artifacts'. Should we be concerned that the interpolation methods (particularly the CNN) are producing upsampled output that contains these harmonic artifacts?*

The artifacts are not ideal but they are certainly expected for the interpolation and conventional downscaling schemes. The CNNs that show the most artifacting are the ones with strictly enforced conservation and no information about the high-resolution climatology, so the presence of the artifacts for these is likely a result of imposing the coarse grid on their outputs. Fortunately, the VSR schemes and CNNs with climatology mostly avoid this effect.

*Sec5: What's the additional computational cost of using VSR methods as opposed to SISR?*

Quite a bit higher, the VSR CNNS have about 25 times the number of parameters, though their computational cost is still miniscule compared to running a high-resolution simulation. We added some more information about this on line 177.

*L114 (and throughout): I don't think chemicals should be in LaTex math mode. If using LaTex, try using the chemformula package and \ch{} command.*

Changed this throughout

*L159: Mention explicitly that SLP is sea level pressure?*

Fixed, and added the units assumed by the re-scaling.

*L415: As far as I can tell, GCM hasn't been introduced.*

Fixed

**Specific Responses to Reviewer 2:**

*(1) Perhaps I misunderstood, but I do not fully grasp how the conservation enforcement layer works. According to equations (3) and (4), the output from this layer appears to be P (mixing ratio in a single low resolution grid cell)? How does this layer interface with the output layer ("0.25° x 0.25° Resolution Output")? Any variable transformations occurring here? For gradient descent to update the parameters, I am under the impression that "weights" and "biases" are needed, but I cannot locate such parameters in this conservation enforcement layer. It would be helpful for readers to have more information.*

I have added some additional exposition to Section 3.4. It is probably easier to check the tracked changes document than list line numbers here. Equation 3 shows the conservation enforcement operator, and the outputs from this layer are f_i. Where f_i is the value in a single high-resolution pixel. f_i takes as input the value of a low-resolution pixel/grid-cell (P) and the values of all the initial CNN output pixels/grid-cells corresponding to P (\mathbf{x}). Equation (4) is simply stating the conservation rule that has been enforced by equation (3). I should clarify that this layer is a simple deterministic function, it has no trainable parameters. It is fairly common in the machine learning literature to also refer to operations within a neural network that do not have trainable parameters as 'layers' (e.g. 'dropout' layer, 'activation' layer, 'noise' layer etc.), and we have added a note to this effect. Because Equation 3 is differentiable, gradients can be propagated through this layer to update weights in the preceding convolutional layers in the network during training.

*(2) I would like to bring to the authors' attention some related prior work that develops a CNN kernel for unstructured grids for spherical signals, which may enhance the model's performance. I believe a short discussion would be beneficial.*

We have added a reference on line 494

*(3) If I'm correct, SISR-CNNs and VSR-CNNs can only process "one pollutant" at a time? Or they can process "multiple pollutants" (such as RGB channels in an image) simultaneously?\\*

As designed in this paper yes. Though it would be trivial to process multiple pollutants at once by treating them as multiple input channels. The conservation enforcement layer and all other modifications we have made from conventional CNN architectures here can be applied in a channel-wise fashion. This makes sense to do if we know that in downstream applications the full set of pollutants will need to be super-resolved together and may even increase the skill of the CNNs. We have added a note about this on lines 503-509 in the discussion section.

*(4) Line 110: There are multiple ways to degrade or prepare low resolution data, making the term "degraded" unclear. Readers may believe, for instance, that the authors conducted simulations with two resolutions separately (using a nested mesh for example). Consequently, I believe that "2-D averaging" should be made explicit here.*

fixed

*(5) Line 171-174: Any citations to support the argument made in factor (1)? Glorot and Bengio (2010) spoke about normalized Initialization for deep feedforward neural networks, but not for convolutional neural networks.*

The main point here, and a motivation of Glorot initialization, is that when layer outputs saturate the following transfer function early in training this leads to near-zero gradients and failed training. This is true of both convolutional and fully connected networks.

*(6) Additional information about the runtime would be beneficial for users. For instance, the number of GPUs used and the training time for SISR-CNNs and VSR-CNNs.*

We have added some information about this on lines 175-176. All models were trained on single GPUs, but this project involved training many models in different configurations so that component was done in parallel.

*Line 190: What is "a-priori"? Is this a typo?*

Yes, removed